# Microstructural Changes in Motor Functional Conversion Disorder: Multimodal Imaging Approach on a Case

**DOI:** 10.3390/brainsci10060385

**Published:** 2020-06-18

**Authors:** Mariachiara Longarzo, Carlo Cavaliere, Giulia Mele, Stefano Tozza, Liberatore Tramontano, Vincenzo Alfano, Marco Aiello, Marco Salvatore, Dario Grossi

**Affiliations:** 1IRCCS SDN, Via Emanuele Gianturco, 113, 80142 Naples, Italy; mariachiara.longarzo@synlab.it (M.L.); giulia.mele@synlab.it (G.M.); liberatore.tramontano@synlab.it (L.T.); vincenzo.alfano@synlab.it (V.A.); marco.aiello@synlab.it (M.A.); direzionescientifica@sdn-napoli.it (M.S.); 2Department of Neuroscience, Reproductive Sciences and Odontostomatology, University of Naples Federico II, 80131 Naples, Italy; stefano.tozza@unina.it; 3Department of Psychology, Università degli Studi della Campania Luigi Vanvitelli, 81100 Caserta, Italy; dario.grossi@unicampania.it

**Keywords:** motor conversion disorder, white matter integrity, functional disorder, functional connectivity, clinical symptoms

## Abstract

Background: Functional motor conversion disorders are characterized by neurological symptoms unrelated to brain structural lesions. The present study was conducted on a woman presenting motor symptoms causing motor dysfunction, using advanced multimodal neuroimaging techniques, electrophysiological and neuropsychological assessment. Methods. The patient underwent fluorodeoxyglucose-positron emission tomography-computed tomography (FDG-PET-CT) and functional magnetic resonance imaging (fMRI) with both task and resting-state paradigms and was compared with 11 healthy matched controls. To test differences in structural parameters, Bayesian comparison was performed. To test differences in functional parameters, a first- and second-level analysis was performed in task fMRI, while a seed-to-seed analysis to evaluate the connections between brain regions and identify intersubject variations was performed in resting-state fMRI. Results. FDG-PET showed two patterns of brain metabolism, involving the cortical and subcortical structures. Regarding the diffusion data, microstructural parameters were altered for U-shape fibers for the hand and feet regions. Resting-state analysis showed hypoconnectivity between the parahippocampal and superior temporal gyrus. Neurophysiological assessment showed no alterations. Finally, an initial cognitive impairment was observed, paralleled by an anxiety and mild depressive state. Conclusions. While we confirmed no structural alterations sustaining this functional motor disorder, we report microstructural changes in sensory–motor integration for both the hand and feet regions that could functionally support clinical manifestations.

## 1. Introduction

Functional neurological disorders (FND) are characterized by neurological symptoms, mainly affecting motor and sensory functions unrelated to brain structural lesions. Motor symptoms include tremor, dystonia, chorea, weakness or paralysis and any type of movement disorder. Altered skin sensation or reduced vision or hearing are additional symptoms. These features are considered conversion signs developed unintentionally in reaction to psychological and/or environmental factors, such as stress or traumatic events [1].

To ascertain the absence of organic lesions, neuroimaging techniques are largely used. In a study by Voon et al. [2], patients with motor conversion tremor exhibited a reduced functional connectivity between the temporo-parietal junction and the sensorimotor and limbic regions during the involuntary tremor but not during the voluntary reproduction of their conversion tremor, suggesting a lack of self-agency as a core feature in conversion disorder. Conversely, Aybek et al. [3] found in patients with motor conversion disorder increased activity in the supplementary motor area (SMA) and in limbic areas associated with emotional processing during motor preparation and inhibition, self-initiated action and sense of agency. Moreover, the authors examining patients with motor conversion disorder observed a greater amygdala and frontal network activity in response to negative emotional stimuli, suggesting an abnormal crosstalk between emotion and motion areas. Several examinations using positron emission tomography (PET) have showed hypermetabolism in the frontal areas [4], explained as the result of active inhibition of the sensorimotor area that translates into limb paralysis; hypermetabolism in the cerebellum and basal ganglia; and an opposite pattern of reduced blood flow in the primary motor cortex [5].

Resting-state studies are lacking in motor FNDs, few studies are reported [6,7,8] and the latter [8] found two connectivity patterns; one showing enhanced connectivity between the right caudate and the left amygdala and bilateral postcentral gyri, and the other showing decreased connectivity between the right inferior parietal cortex and the frontal areas.

The present study was conducted on a woman presenting several motor symptoms causing motor dysfunction, by using a comprehensive approach involving multimodal neuroimaging techniques, such as fluorodeoxyglucose-positron emission tomography (FDG-PET) and functional magnetic resonance imaging (fMRI) both with task and resting states, electrophysiological and neuropsychological assessment. The rationale for the present study is founded on the idea that psychopathological conditions such as functional disorders could be interpreted based on the principle of a close relationship between the brain, mind and body whose manifestations influence each other [9]. Similarly to what was previously observed in hypochondriac patients [10], functional motor conversion disorder is suitable for this interpretation. In particular, the aim was to investigate large-scale brain networks, hypothesizing a higher functional connection between sensorimotor and salience networks.

## 2. Materials and Methods

### 2.1. Case Description

A 55 years-old female patient underwent a neuropsychological evaluation at IRCCS SDN, in order to investigate memory deficit she complained about. She presented motor rigidity (mainly in the lower limbs), lack of balance (she needs the help of a stick to avoid falls), postural instability and lower limb paresthesia. The patient underwent a magnetic resonance imaging of the brain column that showed integrity of the spinal cord. The patient also manifested episodes of spatial disorientation and memory problems. All symptoms begun three years ago, following a verbal and physical aggression she suffered. The patient’s clinical history was characterized by anorexia nervosa when she was 22 years old, as well as post-partum depression and panic disorder of about twenty years. Several traumatic events marked the patients’ life, such as repetitive sexual abuse during infancy and her husband’s arrest two years ago. All these symptoms resulted in a severe impairment to her quality of life, prejudicing functional areas such as work and daily activities. The patient satisfied DSM-5 diagnostic criteria for functional motor conversion disorder. Eleven healthy controls (HC) matched for gender and age were recruited.

### 2.2. Cognitive and Clinical Assessment

The global cognitive status was assessed with the Mini Mental State Examination (MMSE) [11] and the Frontal Assessment Battery (FAB) [12]. The patient performed Raven’s colored progressive matrices [13], Stroop test [14], Rey–Osterrieth Complex Figure [15], and digit span forward and backward (WAIS). We investigated neuropsychiatric symptoms such as depression with the Beck Depression Inventory-II [16], and anxiety through the State-Trait Anxiety Inventory (STAI Y1 and Y2) [17]. We also investigated interoception with the Self-Awareness Questionnaire (SAQ) [18]. Finally, the patient performed the Minnesota Multiphasic Personality Inventory-2 (MMPI-2) in order to investigate personality characteristics.

### 2.3. PET-Computed Tomography (CT) Acquisition and Regional Analysis

Data were acquired on a Discovery IQ hybrid PET-CT scanner (GE Healthcare, Milwaukee, WI, USA), after 4 hours of fasting prior to scan; patient rested in a quiet, dark environment. After the intravenous injection of FDG, the patient rested for 20 min; during the acquisition, the patient lied down supine in the scanner with their arms down and eyes closed. The head was placed naturally so that the patient felt comfortable and motion could be minimized during the acquisition. PET data were acquired using a 3D-mode with a fixed scan duration of 10 min; emission data were corrected for randoms, dead time, scatter, and attenuation from a low-dose CT.

The quantification and reporting of PET/CT images were completed with the Scenium tool of syngo.via software (Siemens Healthineers, Erlangen, Germany), which allows quantitative analysis of PET brain scans by comparison against a suitable normal entries in the database matched for age range, sex and pharmaceutical radiotracer [19]. Brain uptake was normalized for the whole brain instead of the cerebellum in order to include possible variations in metabolism in this region according to neurological evidence.

### 2.4. Magnetic Resonance Imaging Acquisition(MRI)

MRI was acquired after PET-CT examination on a Biograph mMR 3T scanner (Siemens Healthcare, Erlangen, Germany). A 12-channel head coil was used with the following structural and functional sequences:3D T1-Magnetization Prepared Rapid Acquisition Gradient Echo (MPRAGE), voxel size 0.8 × 0.8 × 0.8 mm^3^, Field of View (FOV) 214 × 214 mm, TR/TE/TI = 2400/2.25/1000 ms.Diffusion tensor imaging (DTI), voxel-size 2 × 2 × 2 mm^3^, TR/TE = 3851/84.2 ms, b-value = 1500, 72 directions.Resting-state fMRI, sequence Echo Planar Imaging-Gradient Echo (EPI-GRE), voxel-size 4 × 4 × 4 mm^3^, TR/TE = 1000/21.4 ms, 350 measurements, bandwidth: 2230 Hz.Task fMRI, 3 × 3 × 3 mm^3^, sequence EPI-GRE, TR/TE = 2000/26.4 ms, 15 task measurements and 15 baselines, repeated 5 times. bandwidth: 2005 Hz.

A total of 4 fMRI tasks were performed: finger tapping of the right and left hand, clockwise rotation of the right foot and counterclockwise rotation of the left foot. A task–rest block paradigm was selected, with 30” of carrying out the task and 30” of rest, (repeated 5 times for each sequence). In order to show when the patient was performing the task or rest, a MR-compatible goggles (Nordic Neuro Lab Visual System HD, Bergen Norway) were used to alternately administer two monochromatic backgrounds (green for task, red for rest) through the Opensesame software [20]. The subject was instructed both before and during the examination on the tasks to be performed and when to execute them.

A total of 11 age-matched female HC (55 ± 2.5 years, age range 51–58) performed the same MR acquisition protocol. Informed written consent was obtained both from the patient and HC.

### 2.5. Data Analysis

#### 2.5.1. Structural and Functional Image Processing

For structural image processing, the parcellations of morphological T1-weighted 3D images of HC and the patient were processed with the FreeSurfer v5.1 toolkit [21]. Briefly, this processing includes spatial inhomogeneity correction, non-linear noise reduction, skull-stripping, subcortical segmentation, intensity normalization, surface generation, topology correction, surface inflation, registration to a spherical atlas and thickness calculation [22]. Consequently, the result was normalized by the ratio with the estimated total intracranial volume (eTIV). DTI images were preprocessed with FMRIB Software Library (FSL) (FMRIB, Oxford, UK) and Mrtrix [23] pipelines for denoising, eddy current correction, T1 coregistration and deterministic tractography reconstruction; then the images were uploaded on TrackVis 0.6.1 [24] for virtual dissection. Based on previous tractography work [25], regions of interest (ROI) were defined manually on the axial, coronal and sagittal fractional anisotropy images of each participant, and were used as seed regions for tracking. For the identification of pyramidal tracts, ROI were placed in the medial pons (representing the pontine pyramidal tract) and in the posterior limb of the internal capsule. Regarding U-shape fibers, in order to detect them, the threshold fibers’ turning angle was increased to 90 degrees [26]. All white matter connections projecting between S1 and M1 were captured using an inclusive two-ROI approach. The first ROI was manually traced along the extent of the precentral gyrus (corresponding to M1) in a dorsal-to-ventral manner to incorporate the body of white matter within the gyrus. The second ROI was traced along the extent of the postcentral gyrus (corresponding to S1) in a dorsal-to-ventral manner. All white matter connections projecting between the two ROIs were identified. The connections of the hand region were identified using the anatomical landmark of the inverted omega of the precentral gyrus. The foot tracts were identified as the connections between M1 and S1 located along the most dorsal, and ventral aspect of the central sulcus, respectively (separate to the hand region tract). For all tracts, extraneous or artefactual streamlines were excluded using a not-ROI. Fractional anisotropy (FA), apparent diffusion coefficient (ADC), axial diffusivity (AD), radial diffusivity (RD) values were calculated for each tract.

Regarding functional image processing, task fMRI data were analyzed with functional connectivity toolbox (CONN) [27] and SPM software (Statistical Parametric Mapping: The Analysis of Functional Brain Images). Preprocessing was carried out in SPM using a pipeline that includes realignment, slice-timing and normalization in the Montreal Neurological Institute (MNI) space of functional images.

#### 2.5.2. Statistical Analysis

With structural and microstructural images, in order to test differences in brain volume, cortical thickness, parcels area and diffusion parameters, Bayesian statistical comparison was performed with “Singlebayes” software [28,29].

Regarding functional images, a first-level statistical analysis to assess the activations of the brain areas of each individual task on a qualitative level was carried out. Finally, a subject vs. control group second-level analysis was performed to assess the differences in brain activations. The same preprocessing used for task fMRI was used for resting-state analysis. The first- and second-level analysis was performed with CONN with a seed-to-seed analysis in order to evaluate the connections between brain regions and to identify variations from the patient to HC. A p-value of 0.05 corrected for false discovery rate (FDR) multiple comparison [30] was considered significant for microstructural and functional analysis.

PET images were coregistered and fused to the CT in order to perform brain parcellation in the AAL (automated anatomical labeling) atlas; 114 regions were obtained, and for each the Standard Uptake Value (SUV) mean of the patient and deviation from mean control values were estimated; brain regions are selected as statistical significant with at least four standard deviations from mean control values (Figure 1).

### 2.6. Neurophysiological Assessment

Only the patient underwent standard neurological evaluation and electrophysiological assessment consisting in the motor (MNCS) and sensory (SNCS) nerve conduction study [31], needle electromyography (EMG) [32] and motor-evoked potential (MEP) by use of transcranial magnetic stimulation [33]. Results from neurophysiological evaluation were compared to internal normal values of the center.

The electroencephalogram (EEG) was recorded in 15 min sessions during which the patient was in a psychosensory resting state and alternated between closed and open eyes [34]. A 32-channel brain cap MR with multitrodes was used (Brain Products GmbH, Gilching, Germany). The electrodes were placed according to the 10-5 system [35] and assembled maintaining a skin–electrode impedance of less than 10 kOhm.

## 3. Results

### 3.1. Cognitive and Clinical Assessment

The patient fully cooperated during the assessment. Speech was spontaneous, coherent and congruous but hypophonic. The cognitive exam revealed an initial cognitive impairment, mainly regarding constructional apraxia, visuo-spatial long-term memory domains and global executive functions. Clinical questionnaires showed anxiety as a personality trait and a mild depressive state. The SAQ revealed enhanced interoceptive awareness. The MMPI-2 test was valid, as shown by the L, F and K scales (respectively 58, 57 and 51). The patient did not try to consciously influence her responses and did not show opposition to communicating her emotional problems; she had a tendency to minimize her psychological problems. Among clinical scales, Hs (81) referring hypochondriac symptoms, D (70) to pessimism and feelings of distrust and Hy (76) to anxiety for physical symptoms, were significantly in elevation (Table 1).

### 3.2. Regional PET-CT Analysis

Regional PET analysis showed two opposite patterns of brain metabolism in the FND case. A significant hypermetabolism occurred in the left middle and superior occipital gyrus, left and right cuneus, left cerebellum crus, and left and right superior parietal gyrus. Moreover, the bilateral putamen, right fusiform gyrus, and bilateral superior frontal gyrus were significantly hypometabolic compared to the control group (Table 2).

### 3.3. Structural and Functional Analysis

Bayesian analysis did not show significant differences in volumes, areas and cortical thickness between the patient and HC. Regarding diffusion data from the Bayesian analysis, no differences were found in pyramidal tracts, while the patient showed higher AD (*p* = 0.03) and ADC (*p* = 0.01) in U-shape fibers of the left hand, higher ADC in U-shape fibers of the right hand (*p* = 0.003) and of the left foot (*p* = 0.02), and reduced FA (*p* = 0.03) and higher RD (*p* = 0.001) in the U-shape of the left foot (Figure 2).

Regarding functional analysis, no significant differences occurred in fMRI task analysis between the patient and HC. The analysis of the resting-state paradigm showed a significant hypoconnectivity between left parahippocampal and right superior temporal gyrus (*p* = 0.027), while no significant differences were found using the left parahippocampal gyrus as the seed in a seed-to-voxel analysis (Figure 3).

### 3.4. Neurophysiological Assessment

Neurological evaluation showed that walking was possible only with bilateral support due to difficulty in the toes and heels. Romberg was negative. Cranial nerves were normal, no wasting, strength or sensory (pinprick and vibration) deficits were detected.

Electrophysiological data are summarized in Table 3. Motor and sensory nerve conduction velocities and amplitude of compound motor (CMAP) and sensory action potentials (SAP) were within normal limits. Needle EMG showed no change in motor unit potentials in the anterior tibialis and gastrocnemius muscles, no spontaneous activity was recorded, and recruitment was reduced but with a fluctuating voluntary effort. Upper and lower limb MEPs were normal.

The EEG showed a regular background rhythm. The alpha rhythm was located over the occipito-parietal areas at 10–11 Hz and it was reactive to the opening of eyes. The theta rhythm occurs over the anterior cerebral areas. A low-voltage Beta rhythm was seen over the fronto-central areas at 13–18 Hz.

## 4. Discussion

The present study investigated neurofunctional alterations in a patient with motor FND by employing multimodal neuroimaging techniques. PET-CT data highlighted the opposite patterns of brain metabolism involving the frontal, parietal and occipital lobes whereas resting-state fMRI, which has been used to map brain activity and functional connectivity based on the strength and temporal coherence of neurovascular-coupled hemodynamic signals [36], showed a selective hypoconnectivity between the parahippocampal cortex and temporal gyrus, in absence of any structural findings. Tractography revealed an alteration of U-shape fibers for sensory–motor integration of the hands and feet without any significant difference of pyramidal tracts.

In our patient, the FDG-PET revealed a reduced metabolism in the prefrontal and parietal cortices which spread motor and cognitive information to the cerebellum, well-known for subtending motor actions. According to a meta-analysis on conversion disorders [37], most studies used the single-photon emission-computerized tomography, whereas only one study used PET in two patients with hysterical motor symptoms reporting, in line with our results, hypoactivation of the prefrontal areas [38]. A reduced glucose metabolism was found also in the superior frontal gyrus (SFG) that includes the SMA; the SFG is mainly activated by motor tasks and is related with sensorimotor regions, covering the role of integration of sensorial and motor information. Additionally, it is partly responsible for cognitive functioning. A reduced glucose metabolism in this area could account for both the patient’s motor and cognitive alterations. Our findings are in line with a PET study that reported selective hypoactivation in the frontal and parietal lobes in patients with FND [39]. As the authors suggested, frontal lobe involvement could account for a behavioral or conduct disorder. They elaborated a clinic radiological model of correspondence between brain areas and selective symptoms. According to this, our patient’s paralysis could be addressed by frontal lobe impairment and her abnormal movement by basal ganglia alteration. A mild speech symptom of hypophonia that the patient complained about is attributable to predominantly parietal and frontal lobe alterations [40]. They hypothesized that if there is poor stability on repeated imagery, it is with high probability an FND. Convergent data about the role of the frontal area in conversion disorder come from fMRI results showing that in motor conversion disorders, aberrant activation of the prefrontal area has been related with impaired control of motor execution [41], and that in patients with psychogenic movement disorder, the connectivity between the amygdala and the SMA seems to facilitate the expression of conversion motor representations [42].

Brain metabolism data are interesting because it is possible to speculate that brain structures that normally regulate emotional activation take the leading role of motor functions in order to contain excessive anxiety and to avoid too much suffering. In this sense, the hypoconnectivity between the left parahippocampus cortex and the right superior temporal gyrus is relevant, as both structures are related to the resting-state network and are responsibly for the recovery of personal memory [43]. Coherently with our findings, the abovementioned meta-analysis highlighted functional differences between patients and healthy controls in the temporal area, recognizing its pivotal role in the network addressing emotional trauma elaboration. In support of our hypothesis, the altered metabolism found in prefrontal cortex could also be a brain sign reflecting the suppression of unwanted memories [44].

A noteworthy finding of the present study is the microstructural alterations of U-shape fibers in brain areas where hands and feet are represented according to the motor–sensory homunculus organization. Microstructural parameters of these tracts were altered for hands bilaterally and for the left foot region. The foot area of the motor–sensory homunculus is represented in the paracentral lobule, while other U-shape fibers connecting pre- and post-central regions, relative to hands and mouth areas [25], are represented on the lateral brain surface [40]. Despite these significant alterations, there are several potential caveats for U-shape fiber dissection: (1) the U-shapes of the feet are small in number and not always present in all subjects, as previously reported by other authors [25]; (2) in order to draw the U-shape fibers, it is necessary to modify the conventional diffusion processing pipeline, including a 90-degrees reconstruction angle for the tracts; (3) ROIs drawing to evaluate these fibers requires a time-consuming manual approach; in fact, to date, there are no automatic systems that simplifies the work. Other possible limitations in this study in structural or functional analysis may be due to the exiguous number of subjects or the fMRI task design.

Electrophysiological evaluation was normal when compared to the healthy controls. MEP showed integrity of the corticospinal tract, also observed in previous studies [45,46,47,48]. The integrity of the corticospinal tract, muscles and peripherical nerves, investigated by standard neurophysiological evaluations, suggests that the disorder could be ascribable to an inadequate motor command delivery; these data support the hypothesis of a psychiatric etiology for motor conversion disorder [49]. In particular, Cantello et al. [50] observed that patients affected by conversion disorder who apparently determined their own limb paralysis become persuaded to have not lost motor abilities and recovered instantaneously after a TMS stimulation that produced a motor action.

Since no structural nor task-related functional alterations have been detected in this patient, our interpretation hypothesis supports the idea that microstructural alterations could be considered a brain correlate of functional disorders [51]. The lack of statistically significant results from the fMRI experiment does not rule out the hypothesis of abnormal cross-talk between emotion and motion areas behind the functional paradigm. It is important to mention that the test was performed on a single subject rather than a control group that can generate a wide variability of the activation areas due to both the subjective variability and the specific limitations of the MR methodology (for example, movement and susceptibility artifacts) [52].

Even if white matter investigations in FND are in their infancy [53,54], existing studies observed correlations between FA value alterations and clinical symptoms. Tomic et al. [55] found that patients with a functional disorder such as fixed dystonia presented a global white matter disconnection affecting principally emotional and sensorimotor brain circuits. Ibai Diez et al. [56] in FND patients found that reduced white matter integrity in limbic areas, such as the stria terminalis, uncinate, cingulum and corpus callosum, is correlated with both clinical symptoms and illness duration.

In other neurological conditions, such as vascular diseases and Alzheimer’s dementia, white matter hyperintensity was mainly observed, which is also associated with a more rapid and severe decline of cognitive profile [57] and emotional symptoms such as depression [58]. In our patient, both mild cognitive deficit and depressive symptoms were observed.

It is plausible that fibers that integrate sensory–motor stimuli, receiving continuously altered information due to motor-functioning alterations, have suffered a progressive modification that is reflected in the disruption of microstructural white matter integrity.

These data are compatible with the idea that the patient’s motor conversion is a functional disorder of motor control as a sort of “protective” condition capable of containing mental suffering that the patient struggles to manage.

## Figures and Tables

**Figure 1 brainsci-10-00385-f001:**
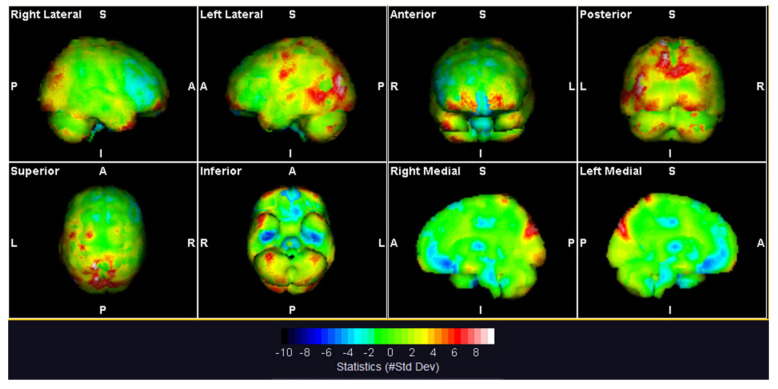
3D visualization of patient positron emission tomography (PET) in comparison with normal database.

**Figure 2 brainsci-10-00385-f002:**
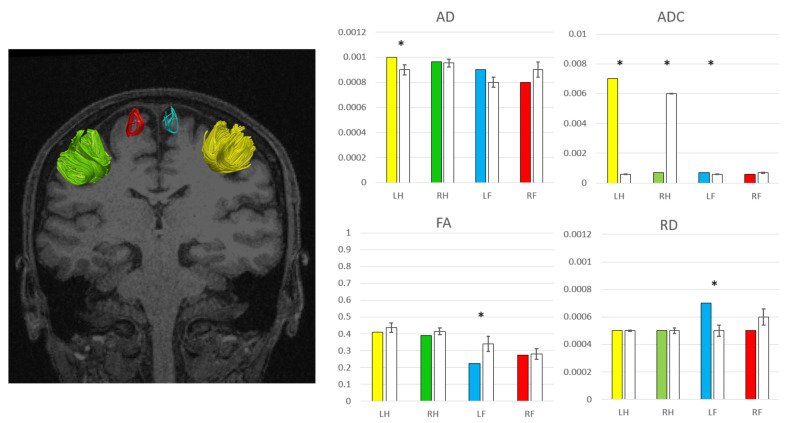
Bilateral hand U-shape fibers: right hand (RH) in green, left hand (LH) in yellow. Bilateral foot U-shape fibers: right foot (RF) in red, left foot (LF) in blue. axial diffusivity (AD), apparent diffusion coefficient (ADC), fractional anisotropy (FA), radial diffusivity (RD) values in the patient (colored bars correspond to the reference U-shapes) and healthy controls (white bars). Significant differences are expressed with “*”.

**Figure 3 brainsci-10-00385-f003:**
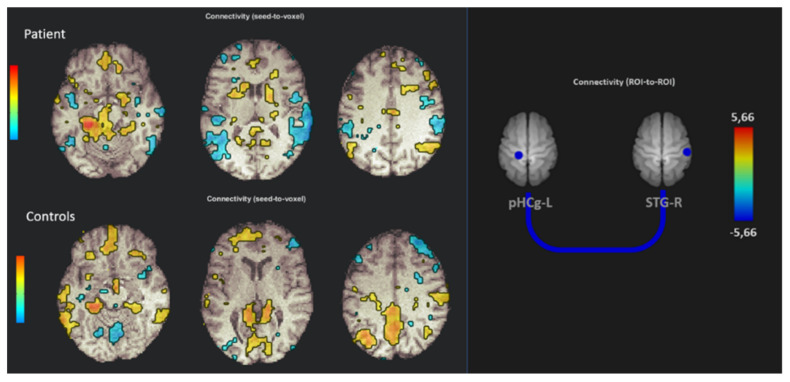
On the left, the resting-state fMRI seed-to-voxel analysis using the left parahippocampal gyrus as the seed in the patient and HC. On the right, seed-to-seed analysis showing significant hypoconnectivity between the left parahippocampal gyrus (pHCg-L) and the right superior temporal gyrus (STG-R).

**Table 1 brainsci-10-00385-t001:** Summary of clinical and cognitive results.

	Raw Score (Cut-off)	Cognitive Domain
MMSE	26 (23)	Global cognitive status
FAB	14 (13.5)	Executive Functions Screening
Ravens’ progressive colored matrices ‘47	23 (18.9)	Fluid intelligence
Stroop test	1: 17” (time) 2: 19” (t) 3: 50” (t) 1 error	Executive functions
Digit Span forward	5	
Digit Span backward	3	Working memory
Rey–Osterrieth Complex Figure Copy Recall	17 3.5	Visuo-spatial long-term memory
BDI-II	17 (14)	Depressive symptoms
STAI Y1 Y2	42 (40) 52 (40)	Behavioral symptoms
SAQ	66	Interoceptive awareness

**Table 2 brainsci-10-00385-t002:** Standard Uptake Value (SUV) mean and deviation from mean database in automated anatomical labeling (AAL) regions (positive deviation for hypermetabolism, negative deviation for hypometabolism).

AAL Region (Left (L) or Right (R))	SUV Mean	Deviation from Database
Superior occipital gyrus (L)	9.2	4.9
Cuneus (L)	10.34	4.5
Middle occipital gyrus (L)	9.33	4.5
Cuneus (R)	10.65	4.3
Cerebellum Crus 2 (L)	8.84	4
Superior parietal gyrus (L)	8.77	4
Superior parietal gyrus (R)	8.32	4
Putamen (L)	8.37	−4.6
Superior frontal gyrus, medialorbital (R)	7.41	−4.6
Fusiform gyrus (R)	6.75	−4.9
Putamen (R)	7.91	−5.4
Superior frontal gyrus, orbital part (L)	6.87	−6

**Table 3 brainsci-10-00385-t003:** Neurophysiological study performed on patient.

Nerve Conduction Study	Parameters	Score (Normal Value)
Median nerve	MNCV (m/s)	60 (≥50)
	dCMAP (mV)	9.2 (≥6)
	F-wave (ms)	25.3 (≤30)
	SNCV (m/s)	56 (≥50)
	SAP (V)	50.5 (≥10)
Tibial nerve	MNCV (m/s)	48 (≥41)
	dCMAP (mV)	23 (≥5)
	F-wave (ms)	44.4 (≤60)
Sural nerve	SNCV (m/s)	50 (≥44)
	SAP (V)	9.8 (≥5)
MEP (upperlimb)	CMCT (ms)	5.3 (≤8)
	Amplitude (mV)	5 (≥2)
MEP (lowerlimb)	CMCT (ms)	12.8 (≤15)
	Amplitude (mV)	2 (≥1.2)

Note: MNCV/SNCV = motor/sensory nerve conduction velocity; dCMAP = distal compound muscular action potential amplitude; SAP = sensory action potential amplitude; MEP = motor evoked potential; CMCT= central motor conduction time.

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
