# Peer review of "Microstructural Changes in Motor Functional Conversion Disorder: Multimodal Imaging Approach on a Case"

_brainsci, 2020, doi:10.3390/brainsci10060385_

Round 1

Reviewer 1 Report

I recommend a major revision of this manuscript to address the following issues:

The scientific value of the manuscript would substantially benefit from the comparison of multimodal images of several patients (instead of only 1 patient) to those of several healthy controls. (Compare this to the works of other authors on motor functional conversion disorder.)

Please have all abbreviations explained.

Please clarify in the methods section: Were electrophysiological examinations conducted with the patient only, or also with healthy controls?

Please discuss: Is the DTI sequence appropriately chosen to detect and depict the U-shaped fibers?

Author Response

The scientific value of the manuscript would substantially benefit from the comparison of multimodal images of several patients (instead of only 1 patient) to those of several healthy controls. (Compare this to the works of other authors on motor functional conversion disorder.)

We thank the reviewer for his suggestion, the discussion has been edited. The comparison of our results with other studies has been implemented in order to give a more complete framework of the state of art about results in motor functional conversion disorder. 

Please have all abbreviations explained.

We are thankful for reviewer comment, we corrected and explained all abbreviations.

Please clarify in the methods section: Were electrophysiological examinations conducted with the patient only, or also with healthy controls?

We thank the reviewer for this comment, electrophysiological evaluation was conducted on patients only. The evaluations results were compared to internal normal values of the center.  We clarified it in “Neurophysiological assessment” section.

Please discuss: Is the DTI sequence appropriately chosen to detect and depict the U-shaped fibers?

We thank the reviewer for the comment. In the DTI processing in order to detect U-shaped fibers, the threshold fibers turning angle wasincreased to90 degrees (default value is 45 degrees)(de Macedo Rodrigues et al. 2012).The size of the ROI placed manually in the S1 and M1 area was chosen in order to detect as many fibers as possible.

Reviewer 2 Report

General comments

The authors have studied the brain metabolism, structure, and function of a patient presenting motor functional conversion disorder and 11 healthy controls using a comprehensive approach (PET, MRI, EEG).  They report a new finding that microstructural changes in patients in sensory-motor integration for both hands and feet regions that could be related to clinical symptoms. The results suggest new insights into the brain microstructure and motor functional conversion disorder.

Resting-state fMRI,

Another important result in this study is the hypoconnectivity between parahippocampal and superior temporal gyrus in resting-state fMRI (Fig. 3). How about the connectivity between other brain regions, could the authors show them in Fig. 3? Or using pHCg-Las the seed, generate the whole-brain correlation map to make the Fig.3 more informative.

Discussion,

Line 314, Even though they did not detect any functional alterations by fMRI in the patient, it may because of the technical limitation of the MRI (e.g. magnetic strength, MRI experiment design et al.) or the number of subjects (this is a case study).

Please discuss the causality between the brain microstructure/ connectivity change and clinical symptoms. Since the symptoms begun three years ago in the patient, Is it possible that the chronic motor functional disorder results in microstructure/ connectivity change?

Presentation,

Overall the manuscript is well written and easy to follow, it still has some room for improvement.

Line 182: please add the color bar to Fig. 1, and indicate the scale of the color bar.

U-shape fiber,

U-shape fiber (microstructure) is highly related to the 2 ROIs which were manually drawn by the researcher. I appreciate that the authors have discussed the potential caveats for the U-shape fiber in the study (Line 293~300).

Statistical Analyses

The authors use a wide range of statistical methods to reach their conclusions.  It is helpful to the reader that the paper has an organized statistical methods section with details.

Suggestion

It will increase the visibility of the study if the authors could reflect the conclusion of the study in the title.

Author Response

Resting-state fMRI, Another important result in this study is the hypoconnectivity between parahippocampal and superior temporal gyrus in resting-state fMRI (Fig. 3). How about the connectivity between other brain regions, could the authors show them in Fig. 3? Or using pHCg-Las the seed, generate the whole-brain correlation map to make the Fig.3 more informative.

 We thank the reviewer for his comment, the connectivity of other brain regions is not showed in figure 3 because they are not statistically significant. We added in figure 3 a seed to voxel whole brain correlation map between the patient and controls using pHCG-L as the seed.

Discussion,Line 314, Even though they did not detect any functional alterations by fMRI in the patient, it may because of the technical limitation of the MRI (e.g. magnetic strength, MRI experiment design et al.) or the number of subjects (this is a case study).

We thank the reviewer for his comment, useful details for understanding the lack of significant differences have been included in the discussion of the revised manuscript: “The lack of statistically significant results carried out from fMRI experiment does not rule out the hypothesis of abnormal cross-talk between emotion and motion areas behind the functional paradigm. It is important to mention that the test was performed on a single subject, compared to a control group that can generate a wide variability of the activation areas due to both the subjective variability and the specific limitations of the MR methodology (for example, movement and susceptibility artifacts)( McGonigle et al, 2000).”

Please discuss the causality between the brain microstructure/ connectivity change and clinical symptoms. Since the symptoms begun three years ago in the patient, Is it possible that the chronic motor functional disorder results in microstructure/ connectivity change?

 We thank the reviewer for this comment that gives us the opportunity to clarify that is more appropriate refer to a relationship between brain microstructure/connectivity changes and clinical symptoms instead of the causality. The microstructural changes described are manifestations of cerebral plasticity plausible expression of renewed correspondence with psychological state. Three years is a sufficient period to induce such modifications.

Presentation,Overall the manuscript is well written and easy to follow, it still has some room for improvement.Line 182: please add the color bar to Fig.1,and indicate the scale of the color bar.

We thank the reviewer for his comment, Figure 1 was updated and the color bar with its own scale was added.

SuggestionIt will increase the visibility of the study if the authors could reflect the conclusion of the study in the title.

We thank the reviewer for his comment. The title has been changed in: Microstructural changes in motor functional conversion disorder: multimodal imaging approach on a case.

Round 2

Reviewer 1 Report

The authors have improved their manuscript, which now appears ready for publication.